# Boosting Vanilla Lightweight Vision Transformers via Re-parameterization

**Zhentao Tan**[1,3][*] **Xiaodan Li**[4,2]**, Yue Wu**[1]**, Qi Chu**[3]**, Le Lu**[2]**, Nenghai Yu**[3]**, Jieping Ye**[1][†]
Alibaba Cloud[1], Alibaba Group[2], University of Science and Technology of China[3],
East China Normal University[4]
`{tanzhentao.tzt, fiona.lxd, matthew.wy}@alibaba-inc.com`
`{le.lu, yejieping.ye}@alibaba-inc.com,{qchu@, ynh@}ustc.edu.cn`

## Abstract

Large-scale Vision Transformers have achieved promising performance on downstream tasks through feature pre-training. However, the performance of vanilla lightweight Vision Transformers (ViTs) is still far from satisfactory compared to that of recent lightweight CNNs or hybrid networks. In this paper, we aim to unlock the potential of vanilla lightweight ViTs by exploring the adaptation of the widely-used re-parameterization technology to ViTs for improving learning ability during training without increasing the inference cost. The main challenge comes from the fact that CNNs perfectly complement with re-parameterization over convolution and batch normalization, while vanilla Transformer architectures are mainly comprised of linear and layer normalization layers. We propose to incorporate the linear ensemble into linear layers by expanding the depth of the linear layers with batch normalization and fusing multiple linear features with hierarchical representation ability through a pyramid structure. We also discover and solve a new transformer-specific distribution rectification problem caused by multi-branch re-parameterization. Finally, we propose our Two-Dimensional Reparameterized Linear module (TDRL) for ViTs. Under the popular self-supervised pre-training and supervised fine-tuning strategy, our TDRL can be used in these two stages to enhance both generic and task-specific representation. Experiments demonstrate that our proposed method not only boosts the performance of vanilla Vit-Tiny on various vision tasks to new state-of-the-art (SOTA) but also shows promising generality ability on other networks. Code will be available.

## 1 Introduction

Inspired by the remarkable success of Transformers in natural language processing (NLP), Vision Transformers (ViTs) (Dosovitskiy et al., 2020) have undergone significant advancements, especially when trained on large-scale datasets with self-supervised learning (e.g., contrastive learning (Chen et al., 2021) and masked image modeling (MIM) (He et al., 2022)). These developments have led to the emergence of large-scale ViTs (Dehghani et al., 2023), which are expected to promote performance mutations similar to Transformers in NLP and eventually become a unified framework for visual or even multimodal tasks (Li et al., 2023; Xu et al., 2023). However, the performance of lightweight ViT models is still far from satisfactory and even inferior to corresponding CNN counterparts (Howard et al., 2019; Woo et al., 2023). Lightweight deep models are still dominated by CNNs or carefully designed hybrid networks (Mehta & Rastegari, 2021; Liu et al., 2023; Vasu et al., 2023b). Given that large ViT models are progressively unifying multimodal feature representation, we believe that it is crucial to explore how to unlock the potential of vanilla lightweight ViTs, thereby achieving uniformity across different model scales.

Fortunately, recent research has recognized this issue and made efforts to take a step forward (Wang et al., 2023; Huang et al., 2023; Ren et al., 2023). MAE-Lite (Wang et al., 2023) gives a detailed

---

[*]This work was supported by Anhui Provincial Science and Technology Major Project (No. 2023z020006) and the National Natural Science Foundation of China (No. 62272430).

[†]Yue Wu and Jieping Ye are corresponding authors.

analysis of the effects of MAE pre-training (He et al., 2022) and finds that ViT-Tiny can achieve promising classification performance with proper configuration: 1) increasing the number of heads to 12; 2) applying attention map distillation during pre-training; 3) carefully adjusting fine-tuning settings. These valuable insights focus on image classification, which may not be applicable to other tasks. TinyMIM (Ren et al., 2023) systematically studies different options in the distillation framework to take full advantage of MIM pretraining. Different from these two aforementioned methods which only focus on applying knowledge transfer during MAE pre-training, G2SD (Huang et al., 2023) proposes to benefit the learning of small/tiny ViTs from both MAE pre-training and downstream fine-tuning. While the generic-to-specific two-stage distillation approach does achieve competitive performance for small models, its applicability may be limited due to the fine-tuning requirements of large-scale teacher models on downstream tasks. Considering that obtaining generic pre-trained large models (He et al., 2022; Radford et al., 2021) is relatively easier, it is more practical to perform knowledge distillation solely during the pre-training stage.

While previous methods focus more on exploring training recipes, we turn to enhance the lightweight ViTs itself: increase the model capacity during training while keeping the inference unchanged via re-parameterization technology (Ding et al., 2021b;a; 2019; 2022). A typical re-parameterized module usually consists of multi-branch networks primarily composed of convolutions and batch normalization (Ioffe & Szegedy, 2015). Thanks to the particularity of batch normalization, these modules retain their adaptive normalization during training and can be merged into a single convolution operation at inference. Consequently, the additional parameters within the re-parameterized module do not increase the inference cost. This approach has been successfully employed in the CNN-related networks (Ding et al., 2021b;a) and is even considered a default technique in recent lightweight network designs (Vasu et al., 2023b;a). However, these convolution-based re-parameterized modules can not be directly applied to vanilla Transformers because of their non-convolution architecture. How to adapt this paradigm to vanilla ViTs remains unexplored.

In this paper, we systematically study the above issues and explore the linear-based re-parameterization of vanilla Vision Transformers, without incorporating any convolutional operations. To enhance the training capacity of the linear layer, we design a delicate linear stack with batch normalization in between them to incorporate adaptive normalization. A pyramid multi-branch structure is further proposed to fuse hierarchical feature representations from linear-based branches of different depths. Additionally, we discover the importance of distribution consistency along the depth dimension of deep networks for training stability, especially for self-attention in ViTs. To alleviate the distribution changes caused by the additive mechanism of re-parameterization, we incorporate additional distribution rectification operations to normalize the outputs. Based on the above designs, we propose a Two-Dimensional Re-parameterized Linear module (TDRL) for ViTs.

To validate the effectiveness of our TDRL, we follow the pre-training and fine-tuning pipeline of MAE He et al. (2022) and apply TDRL to the ViT-Tiny model. It achieves new state-of-the-art performance on various visual tasks, such as image classification, semantic segmentation, and object detection. Further experiments on more models/architectures including the relatively larger ViT-Small network Dosovitskiy et al. (2020), generation network (Ho et al., 2020), and SOTA lightweight networks (Vasu et al., 2023b) also validate the generality of our proposed re-parameterization method.

## 2 RELATED WORKS

**Vision Transformers.** ViTs (Dosovitskiy et al., 2020) have established the dominant position of Transformer architecture in the vision domain recently. It shows competitive performance on various downstream tasks (Touvron et al., 2021; Li et al., 2022; Liu et al., 2021; Zhang et al., 2022a; Peng et al., 2023; Zhai et al., 2022; Zamir et al., 2022; Tan et al., 2023; Zhang et al., 2022b). However, compared to CNN counterparts, ViTs perform poorly in limited model capacity or data scale due to the lack of inductive bias (Dosovitskiy et al., 2020; Touvron et al., 2021). Most lightweight ViT works draw inspiration from CNN designs to build hybrid architecture (Mehta & Rastegari, 2021; Wu et al., 2022b; Chen et al., 2022; Liu et al., 2023; Vasu et al., 2023a), while few works tempt to improve the performance of vanilla ViTs (Wang et al., 2023; Huang et al., 2023; Ren et al., 2023). In this paper, we focus on improving the vanilla lightweight ViT networks via re-parameterization.

**Self-supervised Learning.** Self-supervised learning is the mainstream powerful paradigm for representation modeling without the requirement of data labels (Balestriero et al., 2023). Among them,

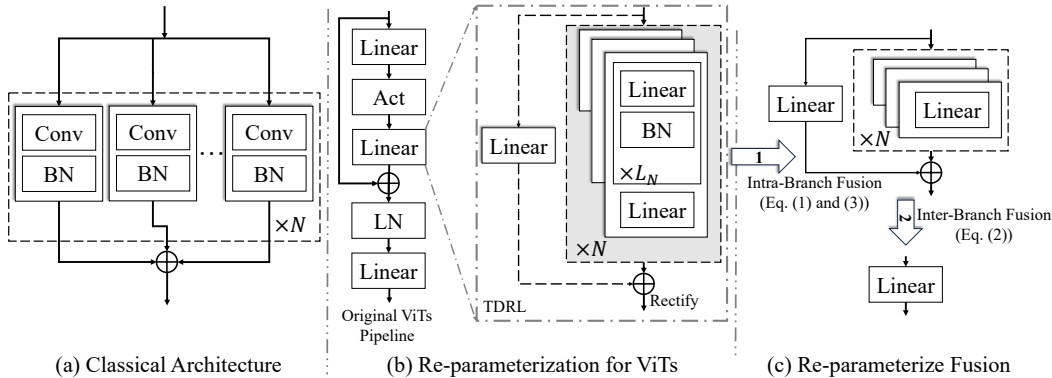

Figure 1: (a) Classical convolution-based re-parameterization architecture. (b) TDRL follows a pyramid design of depths (i.e., $L_n = n, n = \{1, 2, ..., N\}$). Dashed lines with a single linear layer indicate the skip connection. "Rectify" is the distribution rectification. (c) Re-parameterization fusion: 1) merge each rep-branch into a single linear layer through Equation 13; 2) merge multiple branches (each branch contains one linear layer) through Equation 2.

masked image modeling (MIM) has achieved surprising performance on ViTs (He et al., 2022; Bao et al., 2021). Taking raw pixel, semantic features, or discrete tokens as reconstruction targets, most methods explore performance upper bound by finding better supervisions or scaling up model capacity (Dehghani et al., 2023; Wei et al., 2022; Fang et al., 2023; Peng et al., 2022). It has been demonstrated that MIM technologies can benefit large models. But their performance on lightweight ViTs is always overlooked. MAE-Lite (Wang et al., 2023), TinyMIM (Ren et al., 2023), and G2SD (Huang et al., 2023) are recent methods that investigate lightweight ViTs from the perspectives of training configurations and knowledge distillation (Hinton et al., 2015). Differently, we develop a re-parameterized way for lightweight ViTs to take full advantage of MIM pre-training.

**Structural Re-parameterization.** Structural re-parameterization (or over-parameterization) means the technology to scale up the model capacity during training while keeping the inference unchanged. It is very useful to train compact CNN networks (Ding et al., 2021b;a; Guo et al., 2020; Cao et al., 2022; Ding et al., 2019). These modules are built with linear operations (e.g., $K \times K$ convolutions and average pooling) and batch normalization (Ioffe & Szegedy, 2015) to enhance their training representation ability and keep efficient inference speed. However, these methods are designed over convolution which can not be directly applied to convolution-free vision Transformers. In this paper, we extend them to vanilla ViTs without any local convolutions.

## 3 METHODS

### 3.1 RE-VISITING STRUCTURAL RE-PARAMETERIZATION

We first re-visit the re-parameterization of CNN networks (Ding et al., 2021b). As shown in Figure 1 (a), a typical module is a multi-branch additive architecture. Each branch consists of a convolution and a batch normalization. It enhances the learning ability during training and can be merged into a single convolution layer for efficient inference. The merging process can be divided into two steps:

1) *Intra-Branch Fusion*: the fusion of convolution and batch normalization within each branch. Let $\boldsymbol{W} \in \mathbb{R}^{C_{out} \times C_{in} \times K \times K}$ and $\boldsymbol{b} \in \mathbb{R}^{C_{out}}$ denote the weight matrix and bias vector of a convolution layer with $K \times K$ kernel size, $C_{in}$ input channels and $C_{out}$ output channels. The scale, shift, mean, and variance of batch normalization are denoted as $\gamma, \beta, \mu, \sigma \in \mathbb{R}^{C_{out}}$, respectively. The merged convolutional parameters are as follows:

$$\boldsymbol{W}'_{i,:,:,:} = \frac{\gamma_i}{\sigma_i}\boldsymbol{W}_{i,:,:,:}, \quad b'_i = \frac{(b_i - \mu_i)\gamma_i}{\sigma_i} + \beta_i, \tag{1}$$

where $i$ is the output channel index. If converting the convolution to a linear layer, we can also easily fuse its weight and bias with batch normalization through Equation 1.

2) *Inter-Branch Fusion*: the multiple branches can be further merged to a single convolution as:

$$\boldsymbol{W}^{''} = \sum_{n=1}^{N} \boldsymbol{W}^{'n}, \quad \boldsymbol{b}^{''} = \sum_{n=1}^{N} \boldsymbol{b}^{'n}, \tag{2}$$

where $N$ is the number of branches, $\boldsymbol{W}^{'n}$ and $\boldsymbol{b}^{'n}$ are the fused weight and bias of the $n$-th branch.

## 3.2 TWO-DIMENSIONAL RE-PARAMETERIZED LINEAR MODULE

CNN networks perfectly complement with re-parameterization due to the wide usage of batch normalization, for its intriguing calculation transformation characteristics during training and inference, which happens to be the core of re-parameterization. However, ViTs are mainly comprised of linear and layer normalization layers (Ba et al., 2016). Due to the fact that the mean and variance of layer normalization depend on the input during inference, layer normalization cannot be merged with other operations statically as the way of batch normalization. We can only re-parameterize linear layers in transformer networks. In other words, we have to propose a new structure mainly based on linear layers while keeping the re-parameterization structure simple and mergeable in inference by incorporating *Linear Ensemble* to linear layers. What's more, we discover and solve a new transformer-specific *Distribution Rectification* problem with re-parameterization.

**Linear Ensemble**. The basic re-parameterized unit of CNNs instinctively incorporates statistical calculation characteristics with batch normalization. Replicating the unit into multiple branches improves the module capacity. Each branch of convolution incorporates explicit inductive bias through shared local kernels and padding for powerful spatial feature representation within a single layer. However, simple linear replicas suffer the homogeneous problem, with each replica updated with almost the same gradients during training. Thus, we propose to stack linear layers with batch normalization in-between them for the linear ensemble. The rationality is three folds: 1) Linear stacking with batch normalization is similar to MLP [1] which is appropriate to transformers and plays an important role to represent rich intra-token information (Dosovitskiy et al., 2020). Thus, it is inherently appropriate for transformer networks. 2) Although batch normalization is not as suitable as layer normalization for ViTs (Yao et al., 2021), it still can be used in-between layers for the linear ensemble while keeping the original layer normalization unchanged. 3) The stacked linear layers with batch normalization can be fused to a single linear layer. The batch normalization can be fused with a precedent linear layer via Equation 1. Let $\boldsymbol{W}^l \in \mathbb{R}^{C_l \times C_{l-1}}$ and $\boldsymbol{b}^l \in \mathbb{R}^{C_l}$ denote the weight and bias of the $l$-th ($l = 1, 2, 3, ..., L$) merged linear layer in the stack. Two adjacent layers (e.g., $(l+1)$-th and $l$-th) can be merged as:

$$W^{'}_{i,j}(l+1, l) = \sum_{k=1}^{k=C_l} W^{l+1}_{i,k} W^l_{k,j}, \quad b^{'}_i(l+1, l) = \sum_{k=1}^{C_l} b^l_k W^{l+1}_{i,k} + b^{l+1}_i, \tag{3}$$

where $i = 1, 2, 3, ..., C_{l+1}$ and $j = 1, 2, 3, ..., C_{l-1}$ are the channel indexes. Based on Equation 3, we can easily merge a sequence of linear layers of any length into one linear layer. Another effective re-parameterization strategy of CNN-based structure is to use different kernel sizes (e.g., $K \times K$ and $1 \times 1$) to vary the learning patterns of branches. Accordingly, we vary the depth of linear stacks to build a pyramid multi-branch topology. The additive combination of features from these re-parameterization branches exhibits abstract representation ability from shallow to deep. It not only enhances the feature representation but also improves the diversity between branches.

**Distribution Rectification.** Numerous previous works have verified that distribution consistency along the depth dimension is critical to deep networks, with influential works like different initialization and normalization methods (Ioffe & Szegedy, 2015; Ba et al., 2016; He et al., 2015; Kumar, 2017). The above proposed re-parameterization structure will change the distribution between inputs and outputs due to its multi-branch additive mechanism, which will affect the training stability of Vision Transformers, especially for Multi-Head Self Attention (MHSA). The standard self-attention (Dosovitskiy et al., 2020) first calculates the attention map $\boldsymbol{A} \in \mathbb{R}^{M \times M}$ based on the pairwise similarity between query $\boldsymbol{Q} \in \mathbb{R}^{M \times C_k}$ and key $\boldsymbol{K} \in \mathbb{R}^{M \times C_k}$, and then computes a weighted sum over all values $\boldsymbol{V} \in \mathbb{R}^{M \times C_v}$:

$$Attention(\boldsymbol{Q}, \boldsymbol{K}, \boldsymbol{V}) = softmax(\frac{A}{\sqrt{C_k}})\boldsymbol{V}, \quad \boldsymbol{A} = \boldsymbol{Q}\boldsymbol{K}^T. \tag{4}$$

---

[1]MLP stacks two linear layers with GELU and our stack multiple linear layers with batch normalization.

Following Vaswani et al. (2017), assuming that the components of $\boldsymbol{Q}$ and $\boldsymbol{K}$ are independent random variables with mean 0 and variance 1, the elements of attention map $A_{i,j} = \sum_{k=1}^{k=C_k} Q_{i,k} K_{k,j}$ have mean 0 and variance $C_k$. Typical Transformers perform *Scaled Dot-Product Attention* which scale the dot product results by $1/\sqrt{C_k}$ to ensure the final output variance back to 1. However, in case $\boldsymbol{Q}$ and $\boldsymbol{K}$ are re-parameterized as $\boldsymbol{Q}^{'} = \sum_{n=1}^{N_Q} \boldsymbol{Q}^n, \boldsymbol{K}^{'} = \sum_{m=1}^{N_K} \boldsymbol{K}^m$, the distribution of $\boldsymbol{A}^{'}$ changes as follows:

$$A_{i,j}^{'} = \sum_{k=1}^{k=C_k} (\sum_{n=1}^{N_Q} Q_{i,k}^n)(\sum_{m=1}^{N_K} K_{k,j}^m) = \sum_{k=1}^{k=C_k} \sum_{n=1}^{N_Q} \sum_{m=1}^{N_K} Q_{i,k}^n K_{k,j}^m. \tag{5}$$

The re-parameterized distribution changes are amplified through the attention mechanism, where the variance of elements in $A_{i,j}^{'}$ increases to $C_k N_Q N_K$. It will increase the probability of extreme values of $\boldsymbol{A}$ and affecting the stability of training[2] (as shown in Figure3). Considering that attention is fragile to distribution and the distribution of $\boldsymbol{Q}, \boldsymbol{K}$ is much more complicated than the above assumption, we use an additional batch normalization to modulate $\boldsymbol{Q}^{'}, \boldsymbol{K}^{'}$ to rectify the distribution changes. In other components like FFN, this distribution change may also result in the convergence bottleneck. Considering that layer normalization is already used between MHSA and FFN, and the variance change is not as large as in attention calculation[3], we adopt the approach in *Scaled Dot-Product Attention* and re-scale the features with $\sqrt{N}$ rather than normalize it.

**Main Architecture.** As shown in Figure 1 (b), our proposed two-dimensional re-parameterized linear module consists of $N$ re-parameterized branches (denoted as *rep-branch*) and an additional linear layer *skip branch*. The outputs of these branches are fused via element-wise addition. Each *rep-branch* consists of $L_N$ basic re-parameterized units (a linear layer followed by a batch normalization layer) and a final linear layer. Let $f_{n,L_n}(\cdot)$ denote the $n$-th *rep-branch* with $L_n$ basic units, $\boldsymbol{X} \in \mathbb{R}^{M \times C_{in}}$ and $\boldsymbol{Y} \in \mathbb{R}^{M \times C_{out}}$ denote the input and output tensors where $M$ is the sequence length. In this pyramid structure, we set the number of basic units in a *rep-branch* the same as the branch index. The whole calculation of TDRL can be formulated as:

$$\boldsymbol{Y} = Rectify(Linear(\boldsymbol{X}) + \sum_{n=1}^{N} f_{n,L_n}(\boldsymbol{X})), \quad L_n = n, \tag{6}$$

where $Rectify(\cdot)$ is the distribution rectification operation mentioned before. It ensures that each *rep-branch* has different approximation abilities, thereby keeping their feature spaces away from each other. In the following, we use P-W$N$S to denote the detailed configuration of this Pyramid architecture with Width of $N$ *rep-branch* and a *Skip branch*. In addition, we also design a Regular version of TDRL whose Depth per branch is the same for comparison.

$$\boldsymbol{Y} = Rectify(Linear(\boldsymbol{X}) + \sum_{n=1}^{N} f_{n,L}(\boldsymbol{X})). \tag{7}$$

Similarly, we denote this type of TDRL with $N$ *rep-branch*, $L$ basic units per branch, and *skip branch* as R-D$LW$N$S$. Compared to this regular version, we will show that the pyramid structure exhibits better performance and diversity under similar model parameter sizes.

Figure 1 (c) shows the merging of the proposed linear re-parameterization module for inference: 1) **intra-branch fusion**: merge batch normalization within each unit via Equation 1 and merge all linear layers in each *rep-branch* to a single linear layer via Equation 3; 2) **inter-branch fusion**: merge all branches via Equation 2. The proposed TDRL can replace arbitrary linear layers in ViTs.

## 4 EXPERIMENTS

### 4.1 EXPERIMENTAL SETTINGS

**Datasets.** Similar to previous MIM methods (He et al., 2022; Wang et al., 2023; Huang et al., 2023), we pre-train our lightweight ViT models on ImageNet (Deng et al., 2009) which contains about 1.2M

---

[2]When extreme value exists in $\boldsymbol{A}$, the $softmax(\cdot)$ function will scale the elements to close to 0 or 1, which results in the extremely small gradients (since $\partial y/\partial x = y(1-y), y = softmax(x)$).

[3]The variance is scaled to $N$ following the same independent random assumption as before.

Table 1: Comparison with SOTA methods on ImageNet validation. Teachers are used by default for methods with pre-training. *FT* and *P* denote fine-tuning epochs and the size of backbone parameters respectively. †means performing distillation during fine-tuning.

| Method | Network | Teacher | *FT* | *P* (M) | Acc(%) |
|---|---|---|---|---|---|
| Without Pre-training | | | | | |
| MobileNet-v3 (Howard et al., 2019) | CNNs | N/A | 600 | 6 | 75.2 |
| ConvNeXt-V1-F (Liu et al., 2022b) | CNNs | N/A | 600 | 5 | 77.5 |
| VanillaNet-5 (Chen et al., 2023) | CNNs | N/A | 300 | 15.5 | 72.5 |
| MobileViT-S (Mehta & Rastegari, 2021) | Hybrid | N/A | 300 | 6 | 78.3 |
| EfficientViT-M3 (Liu et al., 2023) | Hybrid | N/A | 300 | 7 | 73.4 |
| DeiT-Ti (Touvron et al., 2021) | ViTs | N/A | 300 | 5 | 72.2 |
| Manifold-Ti (Jia et al., 2021) | ViTs | CaiT-S24 | - | 6 | 75.1† |
| MKD-Ti (Liu et al., 2022a) | ViTs | CaiT-S24 | 300 | 6 | 76.4† |
| DeiT-Ti (Touvron et al., 2021) | ViTs | RegNetY | 300 | 6 | 74.5† |
| SSTA-Ti (Wu et al., 2022a) | ViTs | DeiT-S | 300 | 6 | 75.2† |
| ImageNet Pre-training | | | | | |
| DMAE-Ti (Bai et al., 2023) | ViTs | ViT-B | 100 | 6 | 74.9 |
| MAE-Lite (Wang et al., 2023) | ViTs | N/A | 100 | 6 | 76.2 |
| MAE-Ti (He et al., 2022) | ViTs | N/A | 200 | 6 | 75.2 |
| TinyMIM-Ti (Ren et al., 2023) | ViTs | TinyMIM-S | 200 | 6 | 75.8 |
| G2SD-Ti w/o S.D (Huang et al., 2023) | ViTs | ViT-B | 200 | 6 | 76.3 |
| G2SD-Ti (Huang et al., 2023) | ViTs | ViT-B | 200 | 6 | 77.0† |
| TDRL (ours) | ViTs | ViT-B | 200 | 6 | **78.3/78.6†** |
| MAE-Lite (Wang et al., 2023) | ViTs | N/A | 300 | 6 | 78.0 |
| D-MAE-Lite (Wang et al., 2023) | ViTs | ViT-B | 300 | 6 | 78.4 |
| TDRL (ours) | ViTs | ViT-B | 300 | 6 | **78.7/79.1†** |

training images. We validate the performance on downstream tasks including image classification on ImageNet(Deng et al., 2009), semantic image segmentation on ADE20K (Zhou et al., 2019), object detection and instance segmentation on MS COCO (Lin et al., 2014).

**Implementation Details.** We mainly conduct experiments on applying the proposed TDRL to the projection layer of $Q, K, V$[4] in multi-head self-attention (MHSA) and two linear layers in the feed-forward network (FFN). The configuration of TDRL remains consistent across all components. Considering both effectiveness and efficiency, the default recipe of TDRL is set to P-W2S.

Following Wang et al. (2023), we conduct experiments on the classical vanilla ViT-Tiny which only contains about 6M parameters. All blocks are re-parameterized with TDRL. Due to the recent SOTA methods (Wang et al., 2023; Huang et al., 2023) adopting MIM pipeline for taking full advantage of self-supervised learning, we also perform experiments based on them for a fair comparison. In pre-training, we use the MAE pre-trained ViT-Base model as the teacher and perform generic distillation recipes as in Huang et al. (2023). More concretely, the student decoder contains 4 Transformer blocks with an embedding dimension of 128. We align the outputs of the student decoder with the 4-th teacher decoder features including visible and masked patches to transfer generic knowledge. We use a single linear layer to align the features of ViT-Tiny and its teacher model and ignore the class token in loss calculating. For optimization, we use AdamW optimizer (Loshchilov & Hutter, 2017) (with the initial learning rate 2.4e-3, weight decay 0.05, and batch size 4096) to train the model for 300 epochs. Images are randomly resized and cropped with a resolution of 224x224.

For image classification, we fine-tune pre-trained models for 200/300 epochs. For semantic segmentation, we replace the backbone of UperNet (Xiao et al., 2018) with ViT-Tiny and fine-tune the model for 160K iterations. We use the BEiT(Bao et al., 2021) semantic segmentation codebase. The input image resolution is 512x512. For object detection and instance segmentation tasks, we follow

---

[4]The final projection layer in MHSA is ignored since its weight can be merged with that of $V$.

Table 2: Validation of semantic segmentation (ADE20K) and object detection tasks (MS COCO). * means that the resolution is 640x640 as in (Li et al., 2021). †means performing distillation during fine-tuning. ‡means that the results are based on Mask R-CNN (He et al., 2017).

| Method | #Params (M) Seg/Det | Segmentation mIoU | Detection $AP^{bbox}$ | $AP^{mask}$ |
|---|---|---|---|---|
| Swin-T (Liu et al., 2021) | 59.9/47.8 | 44.5 | 46.0‡ | 41.6‡ |
| ConvNeXt-T (Liu et al., 2022b) | 60.0/48.1 | 46.0 | 46.2‡ | 41.7‡ |
| DINO-S (Caron et al., 2021) | 42.0/44.5 | 44.0 | 49.1 | 43.3 |
| iBOT-S (Zhou et al., 2021) | 42.0/44.5 | 45.4 | 49.7 | 44.0 |
| MAE-S (He et al., 2022) | 42.0/44.5 | 41.1/44.9† | 45.3 | 40.8 |
| MAE-Ti (He et al., 2022) | 11.0/27.7 | 36.9/42.0† | 37.9/43.5† | 34.9/39.0† |
| MAE-Lite (Wang et al., 2023) | 11.0/27.7 | 37.6 | 39.9* | 35.4* |
| D-MAE-Lite (Wang et al., 2023) | 11.0/27.7 | 42.0 | 42.3* | 37.4* |
| G2SD-Ti (Huang et al., 2023) | 11.0/27.7 | 41.4/44.5† | 44.0/46.3† | 39.6/41.3† |
| TDRL (ours) | 11.0/27.7 | **42.5/45.2†** | **46.5/47.4†** | **41.5/42.1†** |

the ViTDet (Li et al., 2022) and use the detectron2 (Wu et al., 2019) codebase to train the model with 64 batch size for 100 epochs. The image resolution is 1024x1024. If performing specific distillation in fine-tuning as Huang et al. (2023), the teacher is ViT-Base which achieves 83.6%, 48.1 mIoU and 51.6 $AP^{bbox}$ on image classification, semantic segmentation and object detection. Re-parameterized architecture is used in the fine-tuning stage. More details are provided in the Appendix A.

## 4.2 COMPARISONS WITH SOTA METHODS

**Image Classification.** In Table 1, we summarize the detailed comparison of our TDRL with several types of SOTA methods on image classification, including supervised methods (e.g., MobileNet-v3 (Howard et al., 2019) and ConvNetXt-V1-F (Liu et al., 2022b)), self-supervised methods (e.g., MAE-Ti (He et al., 2022)) and some distillation methods with vanilla ViT-Tiny (e.g., DMAE-Ti (Bai et al., 2023) and G2SD-Ti (Huang et al., 2023)). In general, our TDRL achieves the best classification accuracy under various epoch settings. For example, compared to vanilla ViTs, it outperforms the best-performed one (e.g., G2SD and MAE-Lite) by 1.3% under 200 fine-tuning epochs and by 0.3% under 300 fine-tuning epochs. Compared to carefully designed CNNs or hybrid networks, i.e., MobileViT-S (Mehta & Rastegari, 2021), TDRL achieves 0.5% improvements. We also follow G2SD to perform specific distillation in fine-tuning and find that our performance can be further improved to 79.1%.

**Dense Prediction Tasks.** Except for classification, we demonstrate the advance of pre-trained models for dense prediction tasks, like segmentation, object detection and instance segmentation. As summarized in Table 2, TDRL achieves the best performance for dense prediction compared with other ViT-Tiny-based methods. Concretely, TDRL obtains more than 3.2 mIoU, 3.9 $AP^{bbox}$ and 3.1 $AP^{mask}$ gains compared with MAE-Ti (He et al., 2022) and MAE-Lite (Wang et al., 2023). Compared with G2SD (Huang et al., 2023) which benefits from two-stage knowledge distillation, TDRL achieves slightly better performance in object detection and instance segmentation without knowledge distillation during fine-tuning. What's more, compared with ViT-Small-based and elaborately designed CNNs/hybrid architectures, TDRL shows surprising performance to some extent. It outperforms MAE-S (He et al., 2022) for both two tasks and shows superiority on all metrics (e.g., 0.7 mIoU) compared with Swin-T (Liu et al., 2021) which contains many inductive biases. We obverse that the performance without fine-tuning distillation on ADE20K is not as extraordinary as on MS COCO (i.e., its non-distillation performance on ADE20K is not close to G2SD with specific distillation), which may be caused by the data size. Insufficient data may require a well-learned teacher to guide the training, which results in the new best performance on ADE20K (i.e., 45.2 mIoU). We provide an additional analysis on large-scale ImageNet in the Appendix B.1. Compared to image classification, the improvement gaps from TDRL on dense prediction tasks are considerably larger, which may be attributed to task difficulty. TDRL can benefit better on complex dense prediction tasks than the classification task by improving representation ability.

## 4.3 ABLATION STUDY AND ANALYSIS

In this section, we systematically study the properties of the proposed TDRL. Experiments are mainly conducted on the ImageNet classification task. By default, we fine-tune the model for 100 epochs without teachers for efficiency. We give more experiments and analysis, such as robustness evaluation and efficiency comparison, in the Appendix.

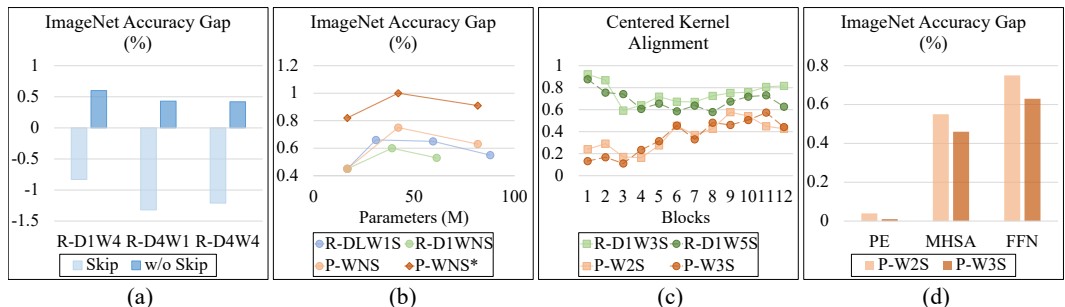

Figure 2: (a) Effects of *skip branch* of TDRL in FFN. (b) Performance of various sizes for three types of TDRL in FFN. The hybrid parameters of these variations are $L$ and $N$. * means that we keep the re-parameterized architecture in fine-tuning. (c) Comparison of CKA (Nguyen et al., 2020) similarity between *rep-branches*. (d) Comparison of embedding positions for ViTs.

**Ablation on Re-parameterization.** Here, we conduct the main ablation of TDRL when applying it to Transformers, including the architecture design and re-parameterized components in ViTs.

1) *Architecture Design.* We first test the importance of *skip branch* through three variants in FFN, involving width and depth expansion. *Skip branch* performs a shorter gradient propagation path compared to *rep-branch* to alleviate gradient vanishing. One can find that all variants suffer from non-negligible performance degradation without it in Figure 2 (a). Then, we validate the superiority of our pyramid structure compared to the regular version in terms of performance and module size. As shown in Figure 2 (b), the pyramid-wise architecture (i.e., P-W$N$S) can achieve the best accuracy compared to other versions (e.g., R-D$L$W1S and R-D1W$N$S) under similar parameter sizes. What's more, we compare the inter-branch diversity between our pyramid structure (P-W$N$S) and the width expansion version (R-D1W$N$S) through CKA similarity in Figure 2 (c). It can be found that P-W$N$S shows a much richer representation ability than R-D1W$N$S, even under a smaller number of *rep-branches*. We also compare the effects with or without TDRL in fine-tuning under P-W$N$S variants and find that keeping it can further stimulate the potential of the lightweight model. Considering both the effectiveness and efficiency, we select P-W2S as the default settings.

2) *Re-parameterized Components.* Finally, we evaluate the effects of applying TDRL to different components of ViTs in Figure 2 (d). Except for Patch Embedding (PE), the introduction of TDRL in other components can bring significant improvements.

**Attention Distribution Rectification.** As analyzed before, re-parameterization of self-attention will amplify the distribution changes, which may seriously affect the training stability. We experimentally track the effects of distribution changes through maximum attention logits and their corresponding attention activation. In detail, we calculate the average maximum activation before and after softmax operation within each block on ImageNet validation datasets (shown in Figure 3 (a)-(b)). Without distribution rectification, the maximum value of their dot product is prone to extreme values for all the 12 Transformer blocks(over $1,000$), leading to attention weights of near-zero entropy (i.e., almost one-hot attention map). In contrast, performing re-scale or normalization prevents divergence due to uncontrolled attention logit growth, which ensures that the behavior of the attention map is similar to that of the baseline. Accordingly, in Figure 3 (c), we can clearly observe a rapid increase in attention logits during the training process[5]. Compared with re-scale and normal-

---

[5]On experiments, we observe the NAN value after a few thousand steps without normalization, which may be caused by the data overflow.

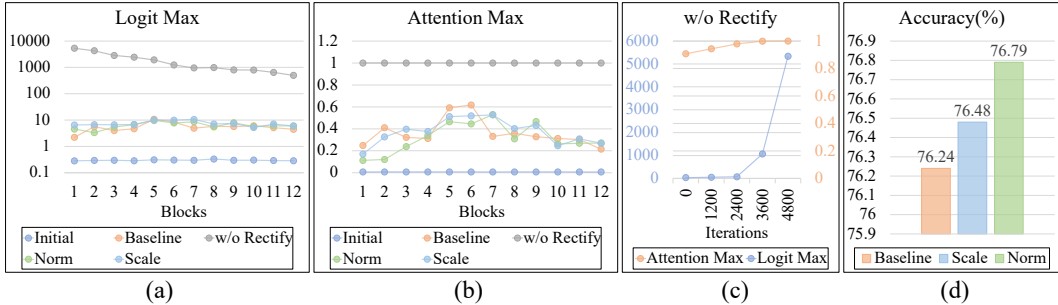

(a)      (b)      (c)      (d)

Figure 3: (a)-(b) Effects of feature distributions on $Q/K$ re-parameterization with the recipe of R-D1W4. The horizontal axis represents the index of blocks. Initial represents the random initialization. The results without rectification come from the last checkpoint before the collapse, while others come from the 300-epoch pre-trained checkpoint. The logit value has been scaled by $1/\sqrt{C_k}$. (c) The trend of logits maximum (blue) and attention maximum (orange) during training. The horizontal axis represents the steps. (d) The Top-1 ImageNet classification accuracy of different settings.

ization, we find that normalization shows superiority in terms of ImageNet classification accuracy (in Figure 3 (d)). We provide more analysis about FFN in the Appendix B.2.

**Generality of TDRL.** In addition to applying TDRL in vanilla ViT-Tiny, we also apply it to other networks to show its generic ability. As summarized in Table 3, we first validate the effect of TDRL in a slightly larger model (i.e., ViT-Small and Swin-Ti (Liu et al., 2021)), then expand the experiments on the recent lightweight CNN, hybrid networks (VanillaNet-5 (Chen et al., 2023) and Mobileone-S0 (Vasu et al., 2023b)), and image generation models (e.g., DDPM (Ho et al., 2020)). It can be found that all these methods benefit from the proposed TDRL, indicating that our proposed TDRL is suitable for various networks on different tasks.

Table 3: Applications of TDRL on various networks and different tasks. For ViT-Small, we follow the same pre-training recipe and fine-tune it for 100 epochs without distillation. The model is re-parameterized in fine-tuning. For other networks, we use official codes and replace the corresponding linear layers with the proposed TDRL. For image generation, we conduct experiments on Cifar10 (Krizhevsky et al., 2009).

| TDRL | Classification Accuracy (%) ↑ | | | | Image Generation FID ↓ |
| | ViT-Small | Swin-Ti | Mobileone-S0 | VanillaNet-5 | DDPM |
|---|---|---|---|---|---|
| ✕ | 80.8 | 76.2 | 71.3 | 71.1 | 10.4 |
| √ | 81.3 (+0.5) | 78.2 (+2.0) | 75.1 (+3.8) | 71.5 (+0.4) | 9.2 (+1.2) |

## 5   CONCLUSION

This paper explores the potential of boosting vanilla lightweight ViTs via re-parameterization. To enhance the representation ability of linear layers in ViTs, we propose a multi-branch pyramid architecture (TDRL) with branches consisting of various depths of linear layers and batch normalization. What's more, we discover and alleviate the distribution explosion problem when applying the proposed TDRL to Vision Transformers by distribution rectification. Experiments show that our TDRL can efficiently improve the performance of lightweight ViTs as well as other transformer or hybrid networks.

**Limitations and societal impact.** Similar to previous re-parameterized methods, our TDRL improves performance without compromising inference efficiency. But it results in extra training costs for larger capacity. However, existing models can still benefit from TDRL under similar training costs (see B.6). We hope our work can promote the research on lightweight ViTs in the future.

## REPRODUCIBILITY STATEMENT

Our proposed method, TDRL, is a lightweight module whose PyTorch-style implementation is provided in the supplementary materials. The pre-training and fine-tuning settings can be found in the Appendix A.1A.2.

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

# A EXPERIMENTAL DETAILS

## A.1 PRE-TRAINING DETAILS

Our self-supervised pre-training strategy follows the recent popular MAE (He et al., 2022), including the optimizer, learning rate, batch size, mask ratio, etc. As for models, we use ViT-Tiny as the encoder and replace its linear layers in MHSA/FFN with the proposed TDRL. By default, we reparameterize all blocks of ViT-Tiny. Following MAE-Lite (Wang et al., 2023), we set the number of heads in ViT-Tiny as 12. In the decoder, we use 4 blocks with an embedding dimension of 128. The teacher model is MAE pre-trained ViT-Base provided by the official repository[6]. We use an additional linear layer to align the last decoder features from the student and the 4-th decoder features from the teacher and calculate the loss for both visible and invisible patches.

## A.2 FINE-TUNING DETAILS

To evaluate the effectiveness of the proposed TDRL, we fine-tune the pre-trained models on three mainstream tasks, including classification, semantic segmentation, object detection and instance segmentation tasks.

Table 4: Fine-tuning settings of ViT-Tiny for ImageNet classification.

| Config | Value (w/o distillation) | Value (w distillation) |
|---|---|---|
| Teacher | N/A | ViT-Base |
| Warmup epochs | {5, 5, 10} | {5, 5, 10} |
| Training epochs | {100, 200, 300} | {100, 200, 300} |
| Layer-wise $lr$ decay | 0.85 (w/o TDRL), 0.65 (TDRL) | 0.75 (w/o TDRL), 0.65 (TDRL) |
| Optimizer | AdamW | |
| Base learning rate | $1e^{-3}$ | |
| Weight decay | 0.05 | |
| Optimizer momentum | $\beta_1, \beta_2 = 0.9, 0.999$ | |
| Batch size | 1024 | |
| Learning rate schedule | Cosine decay | |
| Augmentation | RandAug(10, 0.5) (Cubuk et al., 2020) | |
| Colorjitter | 0.3 | |
| Label smoothing | 0 | |
| Mixup, Cutmix | 0.2, 0 | |
| Drop path | 0 | |

**Image Classification.** We follow previous work (Huang et al., 2023; Wang et al., 2023) to set the fine-tuning recipes and summarize them in Table 4. The difference between using TDRL and merging TDRL in the fine-tuning stage comes from the layer decay (e.g., 0.65 vs. 0.85). To evaluate the effect of these two settings for the baseline, we also fine-tune it with 0.65 layer decay and find that the performance is similar to the original one. Thus, the improvements indeed come from our proposed TDRL, rather than the fine-tuning recipes. When using ViT-Small as the target, the augmentation recipes are the same as Huang et al. (2023).

**Semantic Segmentation.** In this experiment, we use codebase provided by G2SD (Huang et al., 2023) and follow its settings. Differently, we also change the layer decay when using TDRL in fine-tuning. In detail, we set layer decay to 0.80 when using specific distillation, otherwise set it to 0.75.

**Object Detection and Instance Segmentation.** Following G2SD (Huang et al., 2023), we fine-tune the model for 100 epochs with a batch size of 64. The layer decay is set to 0.7 by default. We set the learning rate to $3e^{-4}$ if distillation is not used, otherwise set it to $1e^{-4}$.

---

[6]https://github.com/facebookresearch/mae

### A.3 MORE IMPLEMENTATION DETAILS OF TDRL

When applying TDRL to the Patch Embedding, we should make some modifications. Formally, PE performs $k \times k$ convolution with $k$ stride to encode image patches ( with size of $k \times k$) independently. To combine with the proposed TDRL, we replace the first linear layer of each branch (containing *rep-branch* and *skip branch*) with a convolution layer. At inference time, batch normalization can be converted into convolution followed by Ding et al. (2021b). And we thereafter merge a convolution and a linear as follows:

$$\boldsymbol{W}_{i,j,:,:}^{'} = \sum_{k=1}^{C_{out}^c} W_{i,k}^l \boldsymbol{W}_{k,j,:,:}^c, b_i^{'} = \sum_{k=1}^{C_{out}^c} W_{i,k}^l b_k^c + b_i^l, \tag{8}$$

where $\boldsymbol{W}^c \in \mathbb{R}^{C_{out}^c \times C_{in}^c \times K \times K}$, $\boldsymbol{b}^c \in \mathbb{R}^{C_{out}^c}$, $\boldsymbol{W}^l \in \mathbb{R}^{C_{out}^l \times C_{in}^l}$ and $\boldsymbol{b}^l \in \mathbb{R}^{C_{out}^l}$ are the weights and biases of convolution and linear. And $C_{out}^c = C_{in}^l$ is the prerequisite. By the way, this way can be used when combining our TDRL with other convolutions.

### A.4 MORE DETAILS OF APPLYING TDRL TO OTHER MODELS

Here, we give more details when applying TDRL to different models which are summarized in Table 3. For ViT-Small, we follow the same settings as ViT-Tiny. TDRL is used in both FFN and MHSA. The fine-tuning settings are the same as MAE (He et al., 2022). For Swin-Ti (Liu et al., 2021)[7], we replace the linear layers with the proposed TDRL in the FFN. We train the Swin-Ti with or without our TDRL on ImageNet directly for 100 epochs. For Mobileone (Vasu et al., 2023b)[8] which has already $3 \times 3$ and $1 \times 1$ convolution-based re-parameterization, we replace its $1 \times 1$ convolution-based re-parameterized modules with our TDRL and also combine TDRL with $3 \times 3$ convolution re-parameterized modules. To fuse $3 \times 3$ convolution and linear layer, we can follow the Equation 8. For VanillaNet Chen et al. (2023)[9], we replace its two sequential $1 \times 1$ convolutions with the proposed TDRL. The batch size is set to $512$. For DDPM (Ho et al., 2020)[10], we replace its $1 \times 1$ convolutions within all the self-attention blocks with our TDRL. All the results summarized in Table 3 are reproduced by ourselves. In these experiments, the default setting of TDRL is P-W2S.

## B MORE ANALYSIS

### B.1 DISTILLATION: ONE-STAGE VS. TWO-STAGE

In Table 2, we find that TDRL may still need a well-learned teacher in fine-tuning to achieve the SOTA performance when data is insufficient (e.g., ADE20K (Zhou et al., 2019)), which can be alleviated by increasing the training data (e.g., MS COCO (Lin et al., 2014)). Here, we further compare the one-stage distillation and the two-stage distillation on a large-scale ImageNet classification dataset (Deng et al., 2009) in Table 5. One can see that our proposed TDRL shows superiority compared to the baseline either with specific distillation or without specific distillation. More concretely, the improvement in terms of accuracy is at least larger than $0.97\%$. Compared to the baseline with specific distillation, our TDRL still outperforms $0.62\%$ without the specific distillation, indicating its advantage can be stimulated by enough data.

### B.2 MORE ANALYSIS FOR DISTRIBUTION RECTIFICATION

We have analyzed the impacts of distribution for Attention calculation before. Here, we give more discussion about it for other components of ViTs. Due to the pre-normalization mechanism, the distribution changes of features will accumulate through the skip connection within FFN and MHSA. That is, distribution rectification may also be important when applying TDRL to FFN or the $\boldsymbol{V}$ projection in MHSA. To evaluate it, we test the accuracy gap between the models with and without distribution rectification. As shown in Figure 4, the performance gap increases from around zero to

---

[7]https://github.com/microsoft/Swin-Transformer.git

[8]https://github.com/open-mmlab/mmpretrain

[9]https://github.com/huawei-noah/VanillaNet

[10]https://github.com/zoubohao/DenoisingDiffusionProbabilityModel-ddpm-

Table 5: Comparison on ImageNet with or without specific distillation in fine-tuning. All models are fine-tuned with 100 epochs. Re-parameterized architecture is kept in fine-tuning.

| Method | Specific Distillation | Accuracy (%) | $\triangle$ (%) |
|---|---|---|---|
| Baseline | $\times$ | 76.24 | - |
| TDRL (ours) | $\times$ | 77.39 | +1.15 |
| Baseline | $\checkmark$ | 76.77 | - |
| TDRL (ours) | $\checkmark$ | 77.74 | +0.97 |

$0.18\%$ with the increasing of *rep-branchs* in FFN. It indicates that the greater the change in distribution, the more significant the corrective effect. In addition, we compare the difference between applying TDRL only in FFN and both in FFN and MHSA ($0.13\%$ vs. $0.21\%$). It can be found that the effect of distribution rectification is also proportional to the number of layers applied to TDRL.

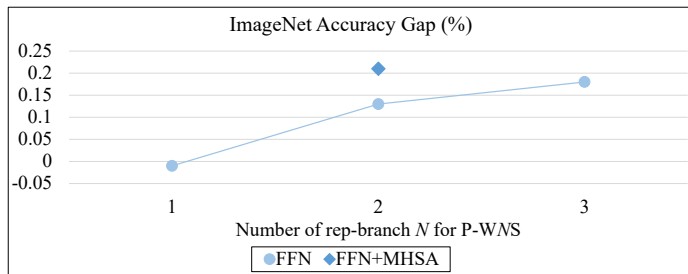

Figure 4: Comparison of distribution rectification. The values are the accuracy gap between the models with and without distribution rectification. We use P-W$NS$ as the configuration of TDRL. All models are fine-tuned for 100 epochs with re-parameterization.

Table 6: Robustness comparison. "IN" is short for ImageNet.

| Method | IN | IN-A | IN-R | IN-S | IN-V2-F | IN-V2-Thr | IN-V2-Top |
|---|---|---|---|---|---|---|---|
| G2SD-Ti | 77.0 | 12.9 | 39.0 | 25.9 | 65.6 | - | - |
| D-MAE-Lite | 78.4 | 13.9 | 40.6 | 28.0 | 66.7 | 74.9 | 80.1 |
| TDRL (ours) | **78.7** | **14.7** | **41.4** | **28.1** | **67.1** | **75.5** | **80.3** |

### B.3 ROBUSTNESS EVALUATION

We evaluate the robustness by directly testing these ImageNet-trained methods on several ImageNet variants, including ImageNet-A (Hendrycks et al., 2021b), ImageNet-R (Hendrycks et al., 2021a), ImageNet-S (Wang et al., 2019) and ImageNet-V2 (Recht et al., 2019). In Table 6, we can see that TDRL outperforms other methods on all test sets, which implies that our method can hold the generalization capability while boosting the downstream task performances.

### B.4 COMPARISON WITHOUT PRETRAINING

Here, we evaluate the efficiency of our proposed TDRL without MIM pertaining. Specifically, we directly train ViT-Tiny and our TDRL on ImageNet for 100 epochs. ViT-Tiny achieves $63.5\%$ accuracy, and our TDRL increases the accuracy to $65.9\%$. This proves that our TDRL does not rely on pretraining and distillation.

Table 7: Comparison of our TDRL under various fine-tuning epochs.

| Fine-tuning Epochs | 100 | 200 | 1,000 |
|---|---|---|---|
| Accuracy (%) | 77.7 | 78.6 | 79.9 |

Table 8: Efficiency comparison between pre-training and inference on V100 GPUs. In the pre-training, the batch size per GPS is set to 256. And the inference batch size is 128. $P$ is the learnable parameters and FLOPs denotes the computational complexity. Values in the ($\cdot$) denote the proportion of increase compared to the baseline (i.e., G2SD-Ti (Huang et al., 2023)).

| Method | Pre-training | | | | Inference Speed |
|---|---|---|---|---|---|
| | Memory (G) | Epoch Times (s) | $P$ (M) | FLOPs (G) | (s/iteration) |
| Baseline | 12.57 | 326 | 5.72 | 1.26 | 0.35 |
| TDRL (ours) | 18.22 (+44.9%) | 462 (+41.7%) | 48.86 | 9.72 | 0.35 (+0%) |

## B.5 LONG EPOCHS OR LARGE MODELS

To evaluate the effect of the training epoch, we fine-tune our TDRL for different epochs. As summarized in Table 7, our proposed TDRL can be beneficial for larger epochs. We further explore the potential of our TDRL for large models, such as DeiT-B (Touvron et al., 2021). In detail, we directly train DeiT-B with or without TDRL on ImageNet for 100 epochs and find that TDRL can still improve the accuracy by $0.6\%$ for large models. Note that we only adopt TDRL in $Q, K, V$ of DeiT-B to reduce training costs. In addition, we find that applying our TDRL to a larger network may require stronger regular constraints (e.g., weight decay) during training.

## B.6 EFFICIENCY COMPARISON

Although TDRL improves the model capacity of ViTs, it brings additional optimized parameters. Here, we summarize the training cost and inference speed in Table 8. It can be found that the cost increase of memory and training times does not exceed $50\%$. And the improvement in training parameters and computational complexity is relatively significant. In the inference stage, our inference speed is as fast as the baseline. For a fair comparison, we reduce the pre-training epochs to keep the total pre-training time the same between the baseline and our TDRL. As summarized in Table 9, our TDRL still outperforms the baseline (i.e., G2SD) by $0.59\%$ in terms of image classification accuracy, which indicates our superiority. In addition, we can flexibly select the recipes of TDRL in terms of module size and replacement places to balance the training cost and test performance. To validate it, we compare the trend of image classification accuracy and pre-training times in Figure 5. As the pre-training cost increases, we can efficiently improve the performance in classification.

## B.7 MORE COMPARISON WITH NAIVE DESIGNS

Here, we compare our proposed TDRL with the naive version that directly converts convolutions to linear layers in Figure 1 (a). As summarized in Table 10, we can find that our proposed TDRL shows better performance than the naive version under similar parameter conditions. For example, TDRL outperforms the naive version by $0.64\%$ at around 30 MB training parameters. It further demonstrates the superiority of our proposed method.

## B.8 OVERFITTING ANALYSIS

As our TDRL is much larger than a single linear layer, one of the concerns of practical application may be listed in the overfitting. In our experiments, we have not observed overfitting issues. The reasons may be as follows: 1) The datasets we used are relatively big for ViT-Tiny, even with our proposed TDRL; 2) Batch normalization in TDRL not only improves the training representation but

Table 9: Comparison of our TDRL and baseline under the similar pre-training time. The configuration of TDRL is P-W1S applied in FFN. The fine-tuning efficiency of the baseline and our method is the same since we merge the re-parameterized architecture after pre-training.

| Method | Pre-train time (hours) | Pre-train epoch | Fine-tune epoch | Accuracy (%) |
|---|---|---|---|---|
| Baseline (G2SD) | 32.33 | 300 | 100 | 76.24 |
| TDRL (ours) | 32.02 | 220 | 100 | 76.73 (+0.59) |

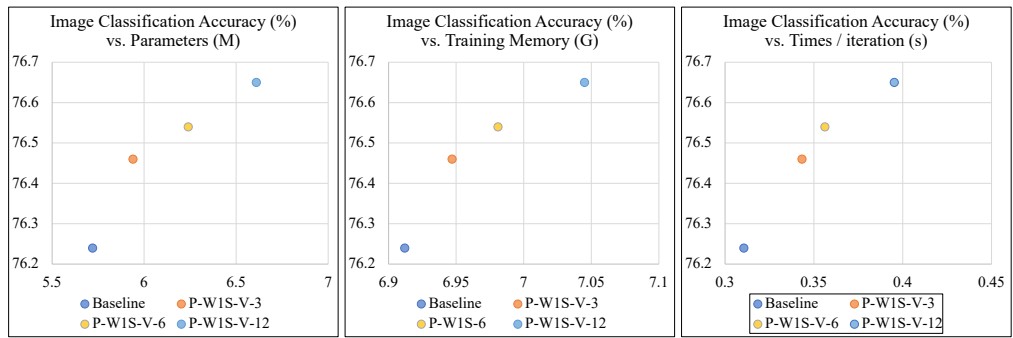

Figure 5: Comparison of image classification accuracy and pre-training efficiency. "-V-$N$" represents that the TDRL is applied in $V$ within MHSA for the first $N$ blocks. The pre-training epoch is set to 300, while all models are fine-tuned for 100 epochs without re-parameterized architectures.

also reduces the risk of overfitting; 3) The feature dimension along the network is not changed (features are limited to the original dimension before outputting from TDRL), resulting in the increased intrinsic dimension. This also reduces the risk of overfitting compared to directly increasing the depth of the network or the feature dimension. We further train ViT-Tiny with or without TDRL on a small dataset (i.e., Cifar10 (Krizhevsky et al., 2009)) which contains 50,000 training images. The results indicate that overfitting still does not occur. TDRL still improves the performance of ViT-Tiny by $1.0\%$ ($70.9\%$ vs. $69.9\%$). When the size of datasets is too small for the network, we may face overfitting issues. However, considering the rapid development of data size and our lightweight model research targets, the probability of overfitting in practical applications is very low.

## B.9   MORE ANALYSIS FOR DENSE PREDICTION TASKS.

To demonstrate the gains resulting from our re-parameterization structure rather than the additional batch normalization layers, we conduct experiments on semantic segmentation that apply batch normalization to the MHSA and FFN. We observe that adding only the batch normalization layer does not bring any effective improvement (37.8 vs. 41.4 in terms of mIoU). This experiment further validates the effectiveness of our re-parameterization structure.

Table 10: Comparison of our TDRL and naive version that directly converts convolutions to linear layers in the typical CNN-based re-parameterized module. The re-parameterized modules are applied in FFN. The naive version contains $8$ branches, while TDRL is set to R-D2W1S for similar parameters.

| Method | Trainin Parameters (M) | Accuracy (%) |
|--------|------------------------|--------------|
| Baseline | 5.72 | 76.24 |
| Naive | 30.75 | 76.26 (+0.02) |
| TDRL | 31.03 | 76.90 (+0.66) |

