# OpenReview forum: "Boosting Vanilla Lightweight Vision Transformers via Re-parameterization"
_ICLR.cc/2024/Conference — ICLR 2024 poster_

### Official Review · Reviewer_Yszy · 2023-10-19

**Soundness:** 4 excellent
**Presentation:** 3 good
**Contribution:** 3 good
**Rating:** 6
**Confidence:** 5

**Summary:**

The paper proposes a new re-parameterization technique for tiny vision transformers. They stack multiple branches with different numbers of linear layers with BN, and the whole branches can be merged into a single linear layer during inference. The experimental results demonstrate the effectiveness of the proposed method.

**Strengths:**

The paper is technically sound and easy to understand. The method of reparameterization on the linear layer is shown to be useful and easy to implement.

The experimental results are rich and show the effectiveness of the proposed method.

**Weaknesses:**

The way of merging a stack of linear layers with BN has already been studied by [1].

You mentioned several times that BN can be used in between layers for nonlinearity, and the proposed re-parameterization method is nonlinear. Why does BN have nonlinearity? Isn't the proposed method a linear ensemble re-parameterization structure?

Why is the experimental setting based on MIM? What is the performance of training ViT with the proposed method on ImageNet directly? What is the performance of training vanilla ViT with MIM?

[1] VanillaNet: the Power of Minimalism in Deep Learning.

**Questions:**

See weaknesses above.

---

> ### Author Response · Authors · 2023-11-17
> **Responses to Reviewer Yszy**
>
> Thanks for your recognition and valuable comments. We respond to your questions about the comparison with VanillaNet, linear ensemble description, and experimental settings in the following. We hope these answers can address all your concerns.
>
> **1. The way of merging a stack of linear layers with BN has already been studied by [1]. [1] VanillaNet: the Power of Minimalism in Deep Learning.**
>
> **Answers:** Thanks for the question. VanillaNet is a great lightweight network that also merges BN and 1x1 Conv (i.e., linear layer) after training for fast inference. Both VanillaNet and our TDRL demonstrate the potential of linear ensemble structure for representation learning. However, the motivation and design of these two methods are different. VanillaNet stacks convolutions and BN like VGG and replaces the commonly used 3x3 kernel with 1x1 to reduce the computational complexity. After training, it merges 1x1 convolutions with BN to speed up the inference speed. In contrast, we want to improve the performance of existing lightweight ViTs. Therefore, we propose to introduce nonlinearity to linear layers by stacking multiple linear layers with BN to improve its representation ability. Furthermore, we propose a pyramid multiple-branch structure to improve both the representation ability and representation diversity. In addition, we analyze the potential training instability issues from both theoretical and experimental perspectives when applying our TDRL in ViTs, and propose a simple and efficient way to solve it. Experiments in Table 1,2,3 show that our proposed TDRL can improve the performance of various lightweight models on different tasks.
>
> **2. You mentioned several times that BN can be used in between layers for nonlinearity, and the proposed re-parameterization method is nonlinear. Why does BN have nonlinearity? Isn't the proposed method a linear ensemble re-parameterization structure?**
>
> **Answers:** Thanks for your comments. In this paper, we write "nonlinear ensemble into linear layers by expanding the depth of the linear layers with batch normalization" in the abstract and the similar introductions in Section 3.2 because though the operation of BN is linear, the mean and variance in BN are calculated by the current features, can not be merged with the adjacent linear layers during training. Thus, we claim that the overall combination is somehow can be regarded as a "nonlinear module". We have updated our revision to make the statement more rigorous.
>
> **3. Why is the experimental setting based on MIM? What is the performance of training ViT with the proposed method on ImageNet directly? What is the performance of training vanilla ViT with MIM?**
>
> **Answers:** Due to that the recent SOTA methods (e.g., G2SD and MAE-Lite) adopt MIM pretraining for taking full advantage of self-supervised learning, we mainly perform experiments based on these pipelines for a fair comparison. Without MIM, we compare the ViT-Tiny models trained on ImageNet directly for 100 epochs and find that our TDRL still achieves better accuracy than the original ViT-Tiny (65.86% vs. 63.47%). We have added this comparison in Section B.4. In Table 1, MAE-Ti is the vanilla ViT-Tiny model with MIM pretraining and supervised fine-tuning. It achieves 75.2% accuracy in ImageNet which is lower than some CNNs or hybrid architectures listed in Table 1.

---

> ### Author Response · Authors · 2023-11-20
> **Official Comment by Authors**
>
> Dear Reviewer Yszy:
>
> We deeply appreciate your valuable comments on our paper. We have provided corresponding responses about VanillaNet and BN, and comparisons without MIM pretraining. It would be great if you could let us know your thoughts on our responses. We hope our response can address your concerns. We would be happy to answer additional questions if any.
>
> Sincerely,  Authors

---

> > ### Comment · Reviewer_Yszy · 2023-11-22
> > **Official Comment by Reviewer yszy**
> >
> > Thanks for the response from the author. I currently keep my score positive.

---

> > > ### Author Response · Authors · 2023-11-22
> > > **Official Comment by Authors**
> > >
> > > Thank you for your recognition and valuable comments.
> > >
> > > Bests,
> > >
> > > Authors

---

### Official Review · Reviewer_GsFu · 2023-10-27

**Soundness:** 3 good
**Presentation:** 4 excellent
**Contribution:** 4 excellent
**Rating:** 8
**Confidence:** 4

**Summary:**

This paper proposes a reparameterization technique to improve performance of lightweight vanilla ViT. The reparameterization technique is used in CNNs but surprisingly unexplored in ViTs. The paper presents a linear-based re-parameterization for vanilla Vision Transformers, without any convolutional operations. The authors introduce a specific linear stack with batch normalization for added nonlinearity and a pyramid multi-branch structure for feature fusion. They also highlight the importance of distribution consistency in deep networks, especially for ViTs, and introduce a Two-Dimensional Re-parameterized Linear module (TDRL) to address this. To enhance re-parameterization, different depths of linear stacks are used to create a pyramid multi-branch structure. Since this multi-branch structure can lead to instability in training,  authors suggest 1) an additional batch normalization to modulate the distributions of Q and K in the attention mechanism to rectify the distribution changes. 2) For other components like Feed Forward Networks (FFN), where the variance change isn't as drastic as in attention calculation, features are re-scaled using the square root of N instead of being normalized.

**Strengths:**

1. The paper looks into reparameterization technique for ViTs performance improvement. The technique that is common in CNN but overlooked in ViTs
2. The proposed TDRL architecture seems to ensure diverse feature spaces among branches, enhancing the model's representation capabilities. TDRL seems to me a modular design that can replace any linear layer in ViTs

**Weaknesses:**

1. The authors experiment on the vanilla ViT-Tiny model. It is not clear what would be the impact of this TDRL module if applied in other tiny transformers such as Swin-tiny etc.
2. The introduction of multiple branches and added nonlinearity might lead to overfitting, especially on smaller datasets. I don't see the authors discussing anything around it

**Questions:**

1. Can you discuss on the computational complexity of the TDRL architecture?

---

> ### Author Response · Authors · 2023-11-17
> **Responses to Reviewer GsFu**
>
> Thanks for your high recognition and valuable comments. In the following, we provide some additional results and discussion about Swin-tiny-TDRL, overfitting issues, and computational complexity.
>
> **1. The authors experiment on the vanilla ViT-Tiny model. It is not clear what would be the impact of this TDRL module if applied in other tiny transformers such as Swin-tiny etc.**
>
> **Answers:** Thanks for the question. The model parameter of Swin-Tiny is 28M, larger than the tiny models (e.g., ViT-Tiny with 6M parameters, and MobileOne-S0 with 2M parameters) used in the paper. So, we don't conduct experiments on it before. When applying TDRL in the FFN components of the Swin-tiny model, we find that the accuracy increases from 76.2% to 78.2% under 100 training epochs, indicating the superiority of TDRL. We have added this comparison in Table 3.
>
> **2. The introduction of multiple branches and added nonlinearity might lead to overfitting, especially on smaller datasets. I don't see the authors discussing anything around it.**
>
> **Answers:** Thanks for the question. Actually, we have not observed overfitting issues in our experiments whose reasons may be as follows: 1) The datasets we used are relatively big for ViT-Tiny, even with our proposed TDRL; 2) Batch normalization in TDRL not only provides nonlinearity during training but also reduces the risk of overfitting; 3) The feature dimension along the network is not changed (features are limited to the original dimension before outputting from TDRL), resulting in the unchanged intrinsic dimension. This also reduces the risk of overfitting compared to directly increasing the depth of the network or the feature dimension. We conduct additional comparison experiments on small cifar10 datasets and find that our TDRL can still improve the performance of ViT-Tiny (70.9% vs. 69.9% for 100 epochs). Of course, when the size of datasets is small for the used network, we may face overfitting issues. However, considering the rapid development of data size and our lightweight model research targets, the probability of overfitting in practical applications is very low. We have added this discussion in Section B.8.
>
> **3. Can you discuss on the computational complexity of the TDRL architecture?**
>
> **Answers:** Thanks. We have added this in Table 8. As for the default settings of TDRL (i.e., P-W2S), the additional FLOPs compared to a single linear is around 5 times. As a result, the whole FLOPs of TDRL is 9.7G, much larger than that of ViT-Tiny (1.3G). Although TDRL increases the computational cost in training, it can be combined for inference without incurring additional costs compared to the original model. In addition, as mentioned in section B.4, we can reduce the overall training costs and computational complexity by reducing the number of replaced linear layers and changing the configuration of TDRL. And experiments demonstrate that under similar training costs, our TDRL still outperforms the vanilla ViT-Tiny.

---

> > ### Comment · Reviewer_GsFu · 2023-11-22
> >
> > Thank you for additional explanation.
> >
> > I will keep my original rating 8 as is.

---

> ### Author Response · Authors · 2023-11-20
> **Official Comment by Authors**
>
> Dear Reviewer GsFu:
>
> We deeply appreciate your insightful comments and suggestions on our paper. We have conducted more experiments on Swin-tiny to demonstrate the efficiency of our method and provided more analysis about overfitting and computational complexity. We hope our response can address your concerns. We would be happy to answer additional questions if any.
>
> Sincerely,  Authors

---

> ### Author Response · Authors · 2023-11-22
> **Official Comment by Authors**
>
> Thank you for your recognition, and for taking the time to review our work and provide constructive feedback.
>
> Bests,
>
> Authors

---

### Official Review · Reviewer_wWZH · 2023-11-03

**Soundness:** 3 good
**Presentation:** 2 fair
**Contribution:** 3 good
**Rating:** 8
**Confidence:** 4

**Summary:**

The paper introduced a new reparametrization approach for ViTs. Unlike standard reparametrization methods only considering convolution layers with kernel sizes larger than 1, this paper studied how to reparametrize the linear layer, the fundamental building block for Vision Transformers. By combining features from multiple branches with different depths, the proposed method exhibited improved performance for ViT-Ti and some other tiny ConvNet-based architectures on ImageNet classification and MSCOCO object detection benchmarks.

**Strengths:**

1. To the best of my knowledge, the proposed method, TDRL, is the first one tailoring for the reparametrization of the linear layer, and thus for ViTs. This is important, particularly for the tiny models, because pure ViTs lag behind the ConvNets and hybrid models due to the lack of inductive bias and inadequate capacity. Reparametrization, training with extra parameters but shrinking back at the inference stage, is a good way to fit more knowledge into the models without incurring extra inference costs.

2. TDRL seems to be a generic method and can be applied to many architectures, e.g., VanillaNet, and different tasks, e.g., DDPM, with linear layers.

3. With the help of varied-depth branches and feature rectification, TDRL achieves promising results on classification and dense prediction tasks, especially the latter.

**Weaknesses:**

1. All the experiments are conducted with tiny or small models and a few pre-training epochs. I am curious whether TDRL would still be beneficial for larger models and when using larger epochs, since reparametrization works for ConvNets with various capacities. This might further strengthen the paper.

2. I noticed that the gain from TDRL is considerably larger on dense prediction tasks. Do the authors have any explanation for it? Is it due to the use of BN in TDRL since some dense prediction tasks favor architectures with BN? If so, can the authors isolate the contributions of reparametrization and BN?

3. Certain parts of the presentation could be further improved. For example, Fig. 1c does not help grasp how to do intra/inter-branch merging.

**Questions:**

See the weakness.

---

> ### Author Response · Authors · 2023-11-17
> **Responses to Reviewer wWZH**
>
> Thank you for your high recognition of our work and valuable comments. Our detailed responses are as follows:
>
> **1. All the experiments are conducted with tiny or small models and a few pre-training epochs. I am curious whether TDRL would still be beneficial for larger models and when using larger epochs, since reparametrization works for ConvNets with various capacities. This might further strengthen the paper.**
>
> **Answers:** Thanks for your valuable comments. We conduct our main experiments with tiny models because our motivation is to boost the performance of lightweight ViTs. To further show its potential for larger models, we apply our TDRL on DeiT-B for 100 training epochs. DeiT-B-TDRL achieves 76.2% accuracy, improving the accuracy of DeiT-B by 0.6%. As for long epochs, we find that our classification accuracy can increase from 78.6% to 79.9% by increasing finetuning epochs from 200 to 1,000. It demonstrates that our TDRL can be beneficial for larger epochs. We have added these comparisons in Section B.5. Due to the limitation of GPUs (e.g., 16 V100 GPUs), we have to spend more time training larger models for larger epochs. For example, we spent around 4 days training TDRL-based DeiT-B for 100 epochs. Thus, we may not be able to provide the results of large models (e.g., DeiT-B) for larger epochs (e.g., 300 epochs) during the author response period. We will add more results in Section B.5.
>
> **2. I noticed that the gain from TDRL is considerably larger on dense prediction tasks. Do the authors have any explanation for it? Is it due to the use of BN in TDRL since some dense prediction tasks favor architectures with BN? If so, can the authors isolate the contributions of reparametrization and BN?**
>
> **Answers:** In our opinion, this may be due to the difficulty of the task. Due to that the dense prediction tasks are more difficult than image classification, the benefits of improving training representation ability through TDRL may be greater. As for BN, it's the core of nonlinearity in TDRL. Without BN, a module composed of multiple linear layers can be equivalent to a single linear layer during training and cannot bring any gain. In the experiment, we observed that when BN was removed from TDRL, its classification accuracy was almost the same as baseline, indicating that a linear re-parameterization structure cannot bring gain. Of course, If there are only BN without heavy parameter structures, ViTs will almost consist of normalization layers and activation layers. Intuitively, this can cause the network to almost lose its classification ability. Thus, isolating the contributions of reparametrization and BN may not be appropriate. We have provided more analysis in Section 4.2.
>
> **3. Certain parts of the presentation could be further improved. For example, Fig. 1c does not help grasp how to do intra/inter-branch merging.**
>
> **Answers:** Thanks for the comment. The entire merging process is obtained through formula derivation (Equation (1), (2), (3)), and it is not easy to provide specific details in Figure 1c. Therefore, we provide the formula numbers involved in each process in Figure 1c to help readers quickly find the corresponding content and understand the entire merging process. We also update the caption of Figure 1 to clarify the target of each merging process.

---

> ### Author Response · Authors · 2023-11-20
> **Official Comment by Authors**
>
> Dear Reviewer wWZH:
>
> We deeply appreciate your insightful comments and suggestions on our paper. We have conducted more experiments for larger models and larger epochs to further demonstrate the potential of our method. In addition, we have provided more analysis and updated our expansion in the paper. We would be happy to answer additional questions if any.
>
> Sincerely,  Authors

---

> > ### Comment · Reviewer_wWZH · 2023-11-22
> > **Response to the authors**
> >
> > Thanks to the authors for their efforts to address the raised concerns.
> > Regarding my second concern, I recommend the authors conduct an extra experiment of the baseline methods with additional BNs (e.g., appended right after Q/K/V or inserted into FFNs) when fine-tuned on the dense prediction tasks. Comparing the baseline with or without BN would also make it somewhat clearer regarding the pure gain from the reparametrization.
> >
> > Despite this concern, I still tend to maintain my original score given the overall strengths of this work.

---

> > > ### Author Response · Authors · 2023-11-22
> > > **Official Comment by Authors**
> > >
> > > Thank you for your recognition and insightful suggestions. We will add this experiment to further explore the pure gain from the reparametrization. Thank you again for helping us improve our paper.
> > >
> > > Best regards,
> > >
> > > Authors

---

### Official Review · Reviewer_WZef · 2023-11-05

**Soundness:** 3 good
**Presentation:** 3 good
**Contribution:** 3 good
**Rating:** 6
**Confidence:** 3

**Summary:**

This paper addresses the problem when using re-parameterization to boost vanilla lightweight ViTs. The main problem is that vanilla Transformer architectures are mainly comprised of linear and layer normalization layers, which do not complement with re-parameterization. This paper incorporates the nonlinear ensemble into linear layers. It also discovers and solves a new transformer-specific distribution rectification problem. It performs lots of experiments in order to verify its effectiveness.

**Strengths:**

The paper is generally well-written, clearly structured, and quite easy to follow.
The target issues of the paper are meaningful and worth exploring. The motivation is clear.
This submission gives a valuable implementation of such an idea.

**Weaknesses:**

1. To some extent, this paper lacks novelty. The intra-branch fusion and inter-branch fusion are commonly seen techniques in the re-parameterization of CNN networks, as shown in Sec 3.1. This paper only applies this technique to ViTs. Although the targeted module is changed from convolutions to linear projections, this modification is straightforward. This paper uses an additional batch normalization to modulate Q^{'} and K^{'} and names this operation a distribution rectification operation. This operation is quite simple.

2. In Table 1, TDRL only outperforms D-MAE-Lite for 0.3 (78.7 vs. 78.4). The advantage does seem not significant.

**Questions:**

See weakness.

---

> ### Author Response · Authors · 2023-11-17
> **Responses to Reviewer WZef**
>
> Thanks for your recognition and valuable comments. In the following, we provide a detailed description of our motivation and the comparison of D-MAE-Lite. We hope our response can address your concerns.
>
> **1. To some extent, this paper lacks novelty. The intra-branch fusion and inter-branch fusion are commonly seen techniques in the re-parameterization of CNN networks, as shown in Sec 3.1. This paper only applies this technique to ViTs. Although the targeted module is changed from convolutions to linear projections, this modification is straightforward. This paper uses an additional batch normalization to modulate Q^{'} and K^{'} and names this operation a distribution rectification operation. This operation is quite simple.**
>
> **Answers:** Thanks for your question. Our motivation is to improve the performance of lightweight vanilla ViT networks through re-parameterization. However, as shown in Table 10, directly applying the recent CNN-based re-parameterization by replacing KxK convolutions with linear projection can not bring efficient improvements. Thus, we propose a nonlinear ensemble, an efficient way to improve the training representation ability by stacking multiple linear layers with batch normalization. We further design a pyramid multi-branch architecture to merge multiple diverse feature representations. In addition, we analyze the potential training instability issues when adopting this multi-branch module to transformers from both theoretical and experimental perspectives. Experiments in Figure 1 show the efficiency of our distribution rectification operation. In summary, although the proposed TDRL is not complex, it can be simply and flexibly adopted in different linear layers in various networks for various tasks (including image classification, semantic segmentation, object detection, and image generation) to improve the performance (shown in Tables 1, 2, 3).
>
> **2. In Table 1, TDRL only outperforms D-MAE-Lite for 0.3 (78.7 vs. 78.4). The advantage does seem not significant.**
>
> **Answers:** Thanks for the question. D-MAE-Lite is a recent SOTA method that focuses on improving the performance of ViT-Tiny, especially for image classification. D-MAE-Lite is pretrained for 400 epochs whose training epoch is larger than ours. When pretrained with 400 epochs, our TDRL outperforms D-MAE-Lite by 0.5%. Considering that the performance difference of some recent methods (e.g., MAE-Ti vs. TinyMIM-Ti and MAE-Lite vs. D-MAE-Lite) on the ImageNet dataset (in Table 1), we believe that this improvement is relatively significant and can demonstrate the superiority of our TDRL.

---

> ### Author Response · Authors · 2023-11-20
> **Official Comment by Authors**
>
> Dear Reviewer WZef:
>
> We deeply appreciate your insightful comments on our paper. In response to your comments, we have further explained the performance issue and provided additional experimental results. It would be great if you could let us know your thoughts on our responses. We would be happy to answer additional questions if any.
>
> Sincerely,  Authors

---

### Official Review · Reviewer_HAf5 · 2023-11-05

**Soundness:** 2 fair
**Presentation:** 2 fair
**Contribution:** 2 fair
**Rating:** 8
**Confidence:** 4

**Summary:**

This paper aims to improve the potential of vanilla lightweight ViTs by exploring the adaptation of the re-parameterization technology to ViTs for improving learning ability during training without increasing the inference cost. The authors propose to incorporate the nonlinear  ensemble into linear layers. They increase the depth of the linear layers during the re-parameterization training stage. And they re-scale the features with $\sqrt(N)$ rather than normalize it in Scaled Dot-Product Attention to solve a transformer-specific distribution rectification.

**Strengths:**

-1- The motivation of this paper is clear.

-2- The pyramid multi-branch topology, i.e., n-th rep-branch with $L_n$ basic units is interesting.

**Weaknesses:**

-1- In the Sec 3.2 Main Architecture, authors use P-WNS to denote the architecture with Width of N rep-branch, what about the different $L_N$ in every rep-branch in the P-WNS? And there is only 1$\times$1 (without K$\times$K) in linear projection, the accuracy gaps in different configurations of "pyramid multi-branch topology" seems minor (Figure 2a).

-2- About the baseline performance. Table 1 reports the performance improvement on ViT-B, with the distillation phase in pre-training. Why the proposed TDRL should be combined with distillation? Why not use a stronger and simpler baseline such as DeiT-B (can obtain 81.8 top-1 accuracy on ImageNet)?

**Questions:**

In Sec 3.2, the authors claim that "The linear operation in transformers focuses more on intra-token semantic mining rather than inter-token spatial exploration. Linear stacking is inherently appropriate to transformer networks." I think the authors should explain this in more details. I am confused why Linear stacking is inherently appropriate to transformer networks.

---

> ### Author Response · Authors · 2023-11-17
> **Responses to Reviewer HAf5**
>
> Thanks for your time and questions. We hope that our responses can address all your questions and concerns and receive your recognition.
>
> **1. What about the different $L_N$ in every rep-branch in the P-WNS? And there is only 1×1 (without K×K) in linear projection, the accuracy gaps in different configurations of "pyramid multi-branch topology" seems minor (Figure 2a).**
>
> **Answers:** $L_N$ denotes the number of basic units (one linear layer with batch normalization) in the rep-branch under N branch re-parameterization. In our pyramid structure, $L_n$ is set to n for n-th rep-branch. Formulaically, $L_n=n, n=1,2,...,N$. We have defined the number of $L_N$ in Equation 6 and in the caption of Figure 1. As mentioned in the introduction, we focus on improving the performance of vanilla lightweight ViTs to achieve uniformity superiority of ViTs across different model scales. Considering that ViTs are non-convolution architectures without KxK convolutions, we thereby design our pyramid multiple re-parameterization architecture to enhance the training representation ability of linear layer, not use KxK convolutions. The results in Figure 2a are not used to show the accuracy gaps in different pyramid multi-branch topologies. Here, we show the importance of the skip branch in the three configurations of the regular version (i.e., P-DLWN), and find that all variants suffer from non-negligible performance degradation (larger than 0.8%) without a skip branch. As for the configurations of P-WNS, we compare their accuracy gaps and model size in Figure 2b. Considering both the accuracy and model size, we chose P-W2S as the default setting.
>
> **2. Why the proposed TDRL should be combined with distillation? Why not use a stronger and simpler baseline such as DeiT-B (can obtain 81.8 top-1 accuracy on ImageNet)?**
>
> **Answers:** In fact, the proposed TDRL does not rely on distillation or MIM pretraining. Due to that the recent SOTA methods (e.g., G2SD and MAE-Lite) adopt distillation and MIM pretraining for taking full advantage of large pre-trained models and self-supervised learning, we perform most experiments under these settings for a fair comparison. When directly training the ViT-Tiny model on ImageNet without MIM pretraining for 100 epochs, our TDRL boosts the performance of ViT-Tiny from 63.47% to 65.86%. It also demonstrates the superiority of our proposed TDRL without distillation and MIM pretraining. As mentioned in the introduction, recent large ViTs have achieved promising performance compared to CNNs, but the performance of vanilla lightweight ViTs is still far from satisfactory compared to that of recent lightweight CNNs or hybrid networks, hindering the uniformity of ViTs across different model scales. Thus, we aim to improve the performance of vanilla lightweight ViTs and not use DeiT-B as the baseline. Also, we find that applying our TDRL on the DeiT-B model can consistently improve the accuracy under 100 training epochs (76.2% vs. 75.6%).
>
> **3. In Sec 3.2, the authors claim that "The linear operation in transformers focuses more on intra-token semantic mining rather than inter-token spatial exploration. Linear stacking is inherently appropriate to transformer networks." I think the authors should explain this in more details. I am confused why Linear stacking is inherently appropriate to transformer networks.**
>
> **Answers:** Thanks for the question. Generally speaking, linear stacking with nonlinearity (batch normalization) is similar to the MLP structure. Since MLP has been proven to be appropriate to transformer networks and plays an important role in learning intra-token representations[1][2][3], we claim that this type of linear stacking is also inherently appropriate to transformer networks. More explanations are as follows:
>
> 1) Linear stacking with batch normalization can be regarded as a non-linear operation in the training stage. In the training stage, this type of rep-branch is somehow similar to the MLP structure in transformers. The differences are: 1) MLP introduces nonlinearity through GeLU, and we implement it through batch normalization; 2) MLP only consists of two linear layers, while our rep-branch contains several linear layers.
>
> 2) MLP  plays an important role in transformer networks to represent rich intra-token information. Many transformers[1][2] have demonstrated the importance of MLP. And another type of network family, namely MLP-Mixer[3], also shows that MLP is very important and MLP-only networks can also achieve promising performance.
>
> Thus, we claim that linear stacking (with batch normalization) is appropriate for ViTs.  In Table 1 and Table 2, we demonstrate that applying our proposed TDRL can improve the performance of transformer networks. We have made our expression clearer in Section 3.2.
>
> [1]. Attention Is All You Need.
>
> [2]. An Image is Worth 16x16 Words: Transformers for Image Recognition at Scale.
>
> [3]. MLP-Mixer: An all-MLP Architecture for Vision.

---

> ### Author Response · Authors · 2023-11-20
> **Official Comment by Authors**
>
> Dear Reviewer HAf5:
>
> We thank you for the precious review time and valuable comments. We have provided corresponding responses and additional experiments, which we hope to address your concerns. We hope to further discuss with you whether or not your concerns have been addressed appropriately. Please let us know if you have additional questions or ideas for improvement.
>
> Sincerely,  Authors

---

> ### Author Response · Authors · 2023-11-21
> **Official Comment by Authors**
>
> Dear Reviewer HAf5:
>
> We greatly appreciate your insightful comment. We have provided a detailed response to your concerns and sincerely hope to discuss it with you. We look forward to hearing from you to help us understand if all your concerns have been adequately addressed. We believe that a collaborative dialogue with you will further strengthen the overall quality of our paper.
>
> Sincerely, Authors

---

> > ### Comment · Reviewer_HAf5 · 2023-11-21
> >
> > Thank you for the author's detailed explanations of $L_n$; they are clear and informative for readers.
> >
> > I'd like to delve further into the results and the notion of "non-linearity" highlighted in the paper's main contribution. I don't aggre with the claim that "Linear stacking with batch normalization can be regarded as a non-linear operation in the training stage," mathematically, these operations are inherently linear. Therefore, I suggest a more comprehensive exploration of the results, especially when considering a "stronger and simpler baseline such as DeiT-B (achieving 81.8 top-1 accuracy on ImageNet)."
> >
> > While the improvement on main papers and "100 training epochs (76.2% vs. 75.6%)" is duly recognized, I am curious about the performance in a more robust setting, specifically with 300 epochs and auto-augmentation. In my own reimplementation of your method, I observed no gains in such conditions.
> >
> > For me, the most intriguing aspect lies in understanding the reasons behind the efficacy of this approach in low baseline scenarios where high baselines might not yield significant improvements. If these insights are substantial, it could elevate the significance of your work beyond showcasing its effectiveness solely in low baseline scenarios, making it more impactful for the vision community.
> >
> > I also respect the scores provided by other reviewers. But I will keep my score for now.

---

> > > ### Author Response · Authors · 2023-11-22
> > > **Official Comment by Authors**
> > >
> > > Dear Reviewer HAf5,
> > >
> > > Thanks for your kind reply and valuable comments. We have updated our expansion about "nonlinearity" and provided more discussion about DeiT-B experiments. We are very eager to know if our response can address your current concerns. Looking forward to your reply.
> > >
> > > Sincerely,  Authors

---

> > > ### Author Response · Authors · 2023-11-23
> > > **Official Comment by Authors**
> > >
> > > Dear Reviewer HAf5,
> > >
> > > Sorry for bothering you again. Since **there is less than a day left until the end of the discussion**, we would like to further discuss with you to address your concerns about BN and DeiT-B.
> > >
> > > 1. As for the expansion of BN, we have already explained in our previous response why the stacking linear layers with BN can be regarded as nonlinearity from a certain perspective (note that we did not claim that BN operation itself is non-linear).
> > >
> > > 2. Regarding the issue you raised about Deit-B, as mentioned in the introduction of the paper, we aim to improve the performance of lightweight ViTs. Therefore, we primarily conducted various experiments to verify the effectiveness of the proposed methods on models such as Tiny-ViT. Although we are committed to meeting your demands and applying the proposed methods to Deit-B, due to the training time required (taking more than 12 days under 16 V100 GPUs), we are unable to provide a response about it during the rebuttal period. However, we have already validated the effectiveness on 100 epochs. For your reproduction, if possible, we sincerely hope that you can provide us with your training details so that we can assist you with the problem you encountered while applying our proposed module to Deit-B.
> > >
> > > In summary, as mentioned by you and Reviewer wWZH, it would make the paper more impactful for the vision community and further strengthen the paper if we have more discussion about the differences between small and large models or TDRL can improve the performance of larger models (e.g., DeiT-B). However, this cannot be a reason to reduce or deny our contributions.
> > >
> > > Sincerely,  Authors

---

> > > > ### Comment · Reviewer_HAf5 · 2023-11-23
> > > >
> > > > Thanks for the authors' reply. I will raise my score to positive. I fully understand the limited rebuttal period and GPU resources. Here I still have some concerns and hope the authors can provide some analyses in final version. For me, I wonder the reason of the accuracy improvement on tiny models, for example, the “non-linear“ perspective, or is just because the increase of the training parameters [1]. In addition, I acknowledge that it's mainly designed for tiny models, I wonder the effectiveness on the DeIT-base models with standard trianing protocal.
> > > >
> > > > [1] Han, Kai, et al. "ParameterNet: Parameters Are All You Need for Large-scale Visual Pretraining of Mobile Networks." arXiv preprint arXiv:2306.14525 (2023).

---

> > > > > ### Author Response · Authors · 2023-11-23
> > > > > **Official Comment by Authors**
> > > > >
> > > > > Dear Reviewer HAf5,
> > > > >
> > > > > Thanks for your recognition and insightful suggestions. We will further explore and analyze the reasons for the role of our method in small models, and mine its potential for larger models.
> > > > >
> > > > > Bests,
> > > > >
> > > > > Authors

---

> ### Author Response · Authors · 2023-11-21
> **Response to Reviewer HAf5**
>
> Dear Reviewer HAf5,
>
> Thanks for your kind reply. Our responses are as follows:
>
> **1. Non-linearity of linear stacking with batch normalizations.**
>
>  **Answer:**  Thanks for your comments. In the previous response, we mentioned "Linear stacking with batch normalization can be regarded as a non-linear operation in the training stage" because though the operation of BN is linear, the mean and variance in BN are calculated by the current batch features, can not be merged with the adjacent linear layers during training. Thus, we claim that the overall combination is somehow can be regarded as a "nonlinear module". Still, this linear stacking is similar to MLP and is appropriate to transformer networks. We also have updated our revision to make the statement more rigorous.
>
> **2. Results on DeiT-B, robust setting, and reimplementation**
>
> **Answer:**  As mentioned in the abstract, we aim to improve the performance of lightweight ViTs. Therefore, our experiments are mostly conducted based on tiny ViTs to investigate whether the proposed method can help boost the performance of lightweight ViTs.
>
> As for the experimental settings, we use official codes like MAE and DeiT, which contain widely used augmentation technologies and training strategies, and we also train the model for 300 epochs (In Table 1). Thus, we believe that our results are obtained in a robust setting.
>
> As for DeiT-B, we use the official code and only use TDRL in Q/K/V for efficiency. We follow the default settings of DeiT-B and only reduce the epoch to 100 due to the training cost and limited time. It still shows the superiority of our TDRL.
>
> As for your reproduction question, could you please provide the training details? Different configurations may lead to very different results. According to the experiment in the paper, the re-parameterization module is not necessarily as large as possible. A larger heavy parameter configuration may increase the difficulty of learning and require changes in the training configuration.
>
> Due to that, the rebuttal stage is drawing to a close, and with our limitations of GPUs, we may not be able to provide the results for 300 epochs right now. But we will add more results in Section B.5 once it's done. We have updated the code for DeiT, you can run the train.sh to train the ViT-Base model.
>
> Sincerely,  Authors

---

### Author Response · Authors · 2023-11-17
**Global response to all reviewers**

We would like to express our sincere gratitude to all the reviewers for their insightful reviews. We thank the recognition and positive ratings given by the majority of the reviewers for our work. To better respond to the reviewer's questions, we have provided separate responses under each reviewer's comments. In addition, we updated the pdf of our paper based on the valuable suggestions of all reviewers as follows. To assist the reviewers in quickly locating their areas of interest, we have highlighted the modifications in red made in the paper.

‒ more experiments about different models, larger epochs, and the DeiT-based training settings;

‒ more analysis of computational complexity, overfitting, and results in Table 1&2;

‒ improve the description of Section 3.2 and Figure 1c.

---

### Public Comment · ~Jack_Ma1 · 2023-11-21
**Experiments without knowledge distillation**

All the experimental results are obtained with knowledge distillation. I cannot figure out whether your core part is useful. Could you please remove KD for experiments?

---

> ### Author Response · Authors · 2023-11-21
> **Responses to Jack**
>
> Dear Jack,
>
> Thanks for your comments. As mentioned in our response to the Reviewers HAf5 and Yszy, our proposed method does not rely on distillation or MIM pertaining. Due to that the recent SOTA methods (e.g., G2SD and MAE-Lite) adopt distillation and MIM pretraining for taking full advantage of large pre-trained models and self-supervised learning, we perform most experiments under these settings for a fair comparison. When directly training the ViT-Tiny model on ImageNet without MIM pretraining for 100 epochs, our TDRL boosts the performance of ViT-Tiny from 63.47% to 65.86% (see in Section B.4). It also demonstrates the superiority of our proposed TDRL without distillation and MIM pertaining. In addition, in Table 3, we show the results of applying our TDRL to different models on different tasks (most of them are not trained with knowledge distillation).
>
>  Sincerely, Authors

---

### Meta-Review · Area_Chair_R1Ux · 2023-12-04

**Metareview:**

This paper aims to improve the performance of tiny VIT models, aligning their performance with those of the similar-size CNN models for computer vision tasks. To this end, the authors propose a new model reparameterization method during the VIT training phase. Specifically, the authors propose to incorporate the nonlinear ensemble into linear layers. Through extensive experiments, the authors find the proposed method can boost the performance of tiny VIT models significantly.

**Justification For Why Not Higher Score:**

- The idea of model re-parameterization is not new. Similar methods (though there is difference) have been explored for CNN models. Thus the value to the community is limited.

- The performance boosting is limited to certain scenarios. The generalization ability of the proposed technique is not clear.

**Justification For Why Not Lower Score:**

- It studies and solves an important problem, i.e., boosting the performance of tiny ViT models.

- The findings and addtiional experiments in the rebuttal may bring some new insight to the community. For example, the proposed method works better for dense prediction tasks, which is interesting.

---

### Decision · Program_Chairs · 2024-01-16

Accept (poster)